# A self-eliminating allelic-drive reverses insecticide resistance in *Drosophila* leaving no transgene in the population

Ankush Auradkar [1,2], Rodrigo M. Corder [3,4], John M. Marshall [4,5] & Ethan Bier [1,2] ✉

Insecticide resistance (IR) poses a significant global challenge to public health and welfare. Here, we develop a locally-acting unitary self-eliminating allelic-drive system, inserted into the *Drosophila melanogaster yellow* (*y*) locus. The drive cassette encodes both Cas9 and a single gRNA to bias inheritance of the favored wild-type (1014 L) allele over the IR (1014 F) variant of the *voltage-gated sodium ion channel* (*vgsc*) target locus. When enduring a fitness cost, this transiently-acting drive can increase the frequency of the wild-type allele to 100%, depending on its seeding ratio, before being eliminated from the population. However, in a fitness-neutral "hover" mode, the drive maintains a constant frequency in the population, completely converting IR alleles to wild-type, even at low initial seeding ratios.

Repeated and prolonged use of a limited spectrum of insecticides has promoted selection for insecticide resistance (IR) in diverse insect populations, undermining standard vector control efforts[1]. The increased prevalence of IR has contributed to an escalation of vector-borne diseases, such as malaria, dengue fever, Chagas disease, sleeping sickness, leishmaniasis, and other trypanosome-mediated diseases[2]. Among these, malaria parasites alone cause >249 million infections and over 600,000 fatalities annually, most being young children[1,3–5]. While the rates of parasite infections have decreased by ~28% from 2000 to 2020[5], predominantly through the use of long-lasting insecticide-treated bed nets (LLINs) and indoor residual spraying, these gains are being eroded due to increasing incidence of IR in insect vectors[6] coupled with the emergence of drug resistance in malarial parasites[7]. The escalation of IR in many crop pests also poses a serious concern, threatening global food security[8,9].

The overuse of insecticides also directly results in ~385 million non-fatal poisonings and ~11,000 deaths annually[10,11]. Furthermore, new insecticides, which typically cost more than $250 million to develop and bring to market[12], cause an indeterminate amount of damage to natural pollinating insect populations[13,14]. Thus, reducing the incidence of IR is a high priority for global health and agriculture.

Mutations in insecticide-target genes are a prevalent cause of resistance in insect species, particularly mosquitoes. Pyrethroids and DDT interact with voltage-gated sodium channel (VGSC), leading to abnormal channel activity, but they cannot bind to or disrupt the function of knockdown-resistant (kdr) mutant ion channels. The most prevalent *kdr* alleles in anophelines mosquito populations comprise L1014F (or equivalently, L995F/S/H in *An. gambiae*), which have independently emerged multiple times across Africa[15].

We recently analyzed the effects of prevalent *kdr* field variants on insecticide susceptibility and fitness in the genetic model *Drosophila melanogaster*[16]. Fly lines carrying the L1014F allele proved highly resistant to DDT, even at high concentrations. A split allelic gene-drive system targeting the 1014F IR allele for conversion to its wild-type (WT) 1014L state demonstrated the feasibility of using such an approach to reverse IR[16]. This split allelic-drive consisted of two separate genetic cassettes: a gRNA drive cassette inserted at the X-linked *yellow* locus and an autosomal static Cas9-expressing source to promote super-Mendelian inheritance of the gRNA element. The driving cassette also carried a second gRNA selectively directing Cas9-mediated cleavage to the 1014F IR allele of the *vgsc* (or *paralytic* = *para* in *Drosophila*) locus, thereby biasing inheritance of the non-cleavable WT insecticide-

[1]Department of Cell and Developmental Biology, University of California San Diego, San Diego, CA, USA. [2]Tata Institute for Genetics and Society, University of California San Diego, San Diego, CA, USA. [3]Department of Parasitology, Institute of Biomedical Science, University of São Paulo, São Paulo, Brazil. [4]Divisions of Biostatistics and Epidemiology - School of Public Health, University of California, Berkeley, CA, USA. [5]Innovative Genomics Institute, Berkeley, CA, USA. ✉e-mail: ebier@ucsd.edu

susceptible *para^{1014L}* allele (Fig. 1a)[17]. In multi-generational cage experiments, this bipartite allelic-drive system reduced the initial frequency of the *para^{1014F}* allele from 83% to 17% over 9 generations, as validated by both DDT resistance assays and deep sequencing[16]. In the current study, we build on this prior system to develop a transiently-acting unitary self-eliminating allelic-drive system (e-Drive) in which all transgenic components are consolidated into a single gene cassette. The e-Drive can fully revert a genetically IR population to wild-type before being eliminated entirely from the population, culminating in a transgene-free endpoint.

## Results

### Constructing a self-eliminating allelic-drive system in *Drosophila*

Based on our prior studies using a split-drive system to revert IR[16], and the well-documented severe male mating fitness cost associated with mutations disrupting function of the *yellow* pigmentation locus[18–23], we developed a self-eliminating drive (e-Drive) by consolidating components from the split-drive into a single gene cassette (Fig. 1a, b). We modified the prior gRNA carrying element by inserting a *vasa-Cas9* encoding transgene, but omitting gRNA-y that mediated copying of the split-drive cassette. We inserted this e-Drive cassette into the *y* locus at the same gRNA-y target site as the prior split-drive construct. A minor optimizing modification was also made to gRNA-F (Material and Methods) that directs selective cleavage of the *para^{1014F}* kdr allele, but not the WT *para^{1014L}* allele. Thus, the unitary e-Drive carries transgenes encoding vasa-Cas9, gRNA-F, and the DsRed marker within a single gene cassette that is inherited in a standard Mendelian sex-linked fashion at the *y* locus.

### The e-Drive efficiently converts the *para^{1014F}* kdr allele to wild-type

We initially characterized performance of the unitary e-Drive element by evaluating its cleavage and mutagenic activities. We crossed flies carrying the e-Drive construct to a strain carrying the insecticide-resistant *para^{1014F}* allele, employing either e-Drive bearing males or females as $F_0$ parents. Genomic DNA extracted from $F_1$ trans-

heterozygous *e-Drive*/+, *para^{1014L}*/*para^{1014F}* females (designated $F_1$ master females) was then assessed for gRNA-F cleavage activity in somatic cells by next-generation sequencing (NGS). We observed that ~80% of the target *para^{1014F}* alleles were mutated in these $F_1$ master females whether the e-Drive originated from their $F_0$ fathers ($F_0$-Male) or mothers ($F_0$-Female) (Supplementary Fig. 1a, b).

We tested the germline efficiency of the e-Drive system by performing two-generation outcrosses and quantifying WT (*para^{1014L}*) versus IR (*para^{1014F}*) allelic frequencies in the $F_2$ offspring. In these experiments, we combined a dominant *mini-white* (mW) marker gene that is linked tightly (~0.5 cM/60 Kb) in *cis* to the *para^{1014L}* allele on the donor chromosome to distinguish it from the receiver ($w^-$) chromosome carrying the *para^{1014F}* kdr target allele (Fig. 2a). Previous studies revealed that transmission of gene-drive elements through the female germline can generate more frequent target-site mutations during early embryogenesis through the error-prone nonhomologous end-joining (NHEJ) DNA repair pathway[24,25]. Thus, we performed reciprocal crosses either by mating $F_0$ males carrying the *e-Drive, $w^-$; mW, para^{1014L}* donor chromosome to $F_0$ females harboring the $y^+$, $w^-$; *para^{1014F}* receiver chromosome or conversely, by crossing $F_0$ donor chromosome females to $F_0$ males carrying the receiver chromosome (Fig. 2b). In both cases, $F_1$ master females were outcrossed in triplicate to *w1118* $F_1$ males, and allelic conversion efficiencies were then quantified in $F_2$ progeny inheriting the $w^-$ receiver chromosome ($w^-$) by NGS sequencing of a PCR fragment spanning the *para^{1014F}* cleavage target site (Fig. 2b).

In experiments where the $F_0$ male carried the e-Drive, we inferred an allelic receiver conversion rate of ~28% in $w^-$ $F_2$ female progeny (based on the assumption that 50% of sequences are derived from the reference $F_1$ *w1118* male parent allele) and a ~40% conversion rate when the e-Drive originated from $F_0$ females (Fig. 2c, Supplementary Fig. 2a). These estimated conversion frequencies are similar to those for the previously analyzed split allelic-drive (~30%)[16]. NGS sequencing also revealed a modest frequency (16–24%) of NHEJ-induced mutant alleles.

In contrast to the analysis of allelic conversion rates in heterozygous female $F_2$ progeny, a substantially higher apparent conversion rate was observed in hemizygous $w^-$ $F_2$ males, which carry only a single X-chromosome. Here, depending on whether $F_0$ grandfathers or

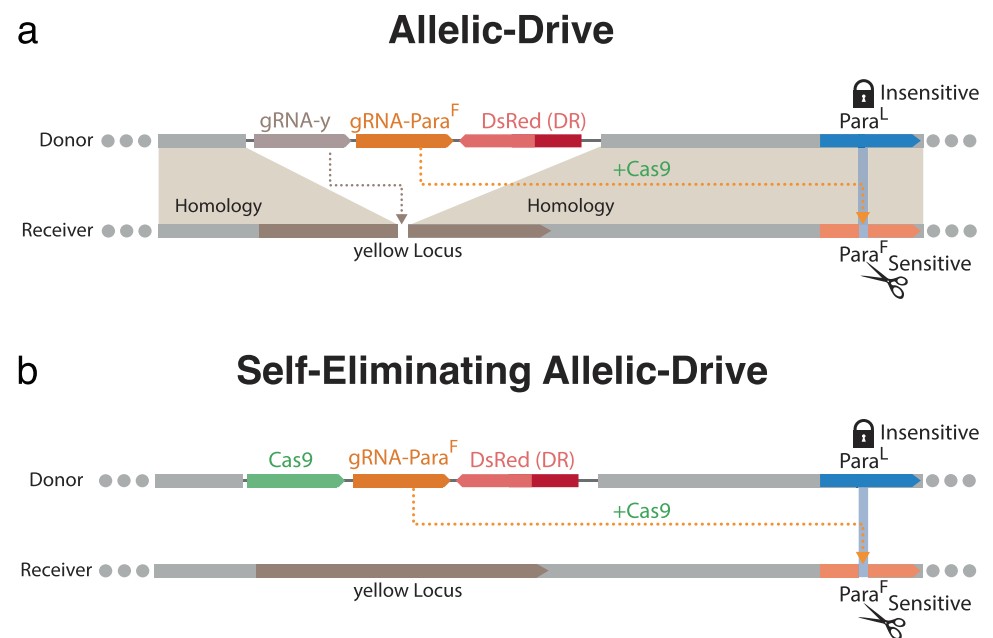

**Fig. 1 | Schemes for split versus unitary self-eliminating *IR* reversal drives.**
**a** Split allelic-drive inserted into the *yellow* (*y*) locus. The drive cassette carries two gRNAs, one to drive itself at *y* (gRNA-y), and the other (gRNA-F) to drive the preferred WT 1014L allele of *para* (by preferentially cutting the undesired IR 1014F allele) using a separately encoded Cas9 cassette. **b** Unitary self-eliminating allelic-drive in *y* locus (e-Drive). The e-Drive (e-Dr gene cassette includes a Cas9 source but lacks gRNA-y and thus does not copy the gene cassette.

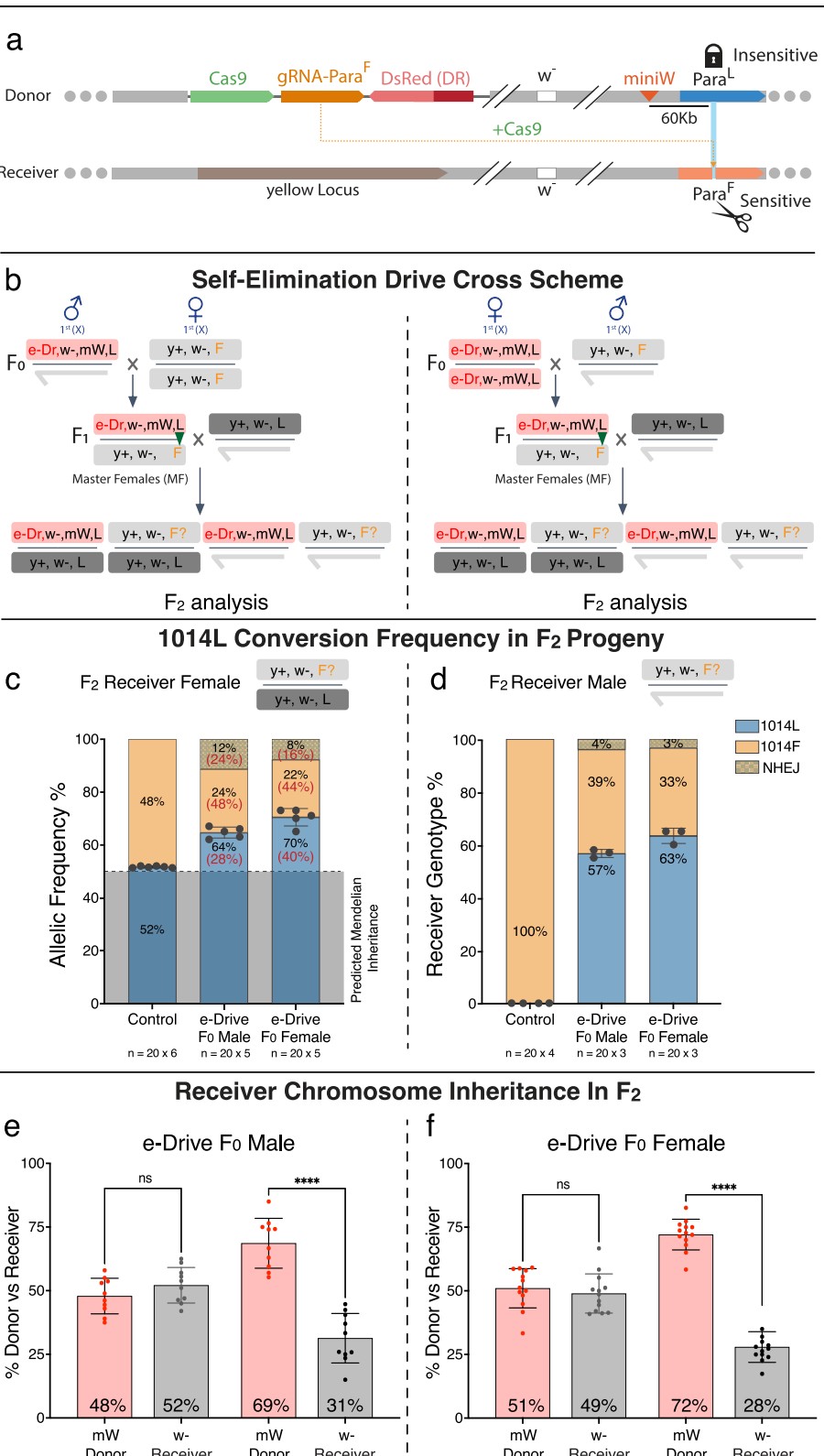

**Self-Elimination Drive Cross Scheme**

**1014L Conversion Frequency in F₂ Progeny**

**Receiver Chromosome Inheritance In F₂**

grandmothers carried the e-Drive, rates of converting the target $w^-$ chromosome from $para^{1014F}$ to $para^{1014L}$ were respectively: 57% for progeny derived from F₀ grandfathers, and 63% for offspring from F₀ grandmothers, with virtually no NHEJ alleles being generated (indeed, we have never recovered any viable *para* mutations with gRNA-F) (Fig. 2d, Supplementary Fig. 2b). The nearly two-fold difference in estimated conversion rates in male (57–63%) versus female (28–40%)

F₂ progeny most likely can be attributed to NHEJ alleles being lethal in males, but not in females who have two X-chromosomes. This factor results in fewer receiver than donor chromosomes being recovered in male versus female offspring (~2:1), an effect that is similar in crosses using either F₀ males or females that transmit the e-Drive cassette. The sizeable deficit of receiver chromosomes recovered in male F₂ progeny suggests that a fraction of these target chromosomes (~50%) were

**Fig. 2 | Assessing allelic-drive conversion efficiency. a** Scoring donor versus receiver chromosomes. The donor chromosome carries the e-Drive (e-Dr) element (Cas9, DsRed and gRNA-F), a *white*[-] allele (*w*[1118]) at the endogenous *w* locus and a *mini-white* marker (*mW*, red triangle) that is linked closely, 0.5 cM (~60 kb), to the uncleavable *para*[WT] donor allele (*ParaL* or *L*, lock icon). The *w*[1118] receiver chromosome carries the cut-sensitive *para*[1014F] allele (*ParaF* or *F*) but lacks the *mW* insertion. **b** Outline of the genetic cross schemes used to assess the allelic conversion efficiency of the e-Drive, when it is introduced by either males (F$_0$, right panel), or females (F$_0$, left panel). F$_1$ master females (heterozygous for the donor *e-Dr,w*[-]*,mW,para*[WT] chromosome and receiver *y*[+]*,w*[-]*,para*[L1014F] chromosome) were crossed individually to *w*[1118] individuals to assess germline receiver allelic conversion in F$_2$ progeny. The conversion event at the *para*[L1014F] locus (*F*) is depicted by a triangle in F$_1$ individuals. **c, d** *para* locus allelic frequencies (1014L, 1014F, and NHEJ) of F$_2$ females (**c**) or males (**d**) inheriting a receiver chromosome (selected for *w*[-] eye phenotype) were determined by sequencing. These F$_2$ progeny, which were collected from crosses originating from F$_0$ male/female grandparents, carried no

e-Drive element (control, left bar), or carried the e-Drive inherited from F$_0$ grandfather (center bar) or from F$_0$ grandmothers (right bar). Error bars indicate mean ± standard deviation, with the mean value indicated in the bar. Gray shaded region represents percentage of 1014L allele (~50%) that is predicted to be derived from the reference F$_1$ male parent allele (thus, in panel **c**, the inferred conversion percentage in females with two X-chromosomes, shown in red type within parentheses and inside the blue boxes, is estimated by the formula: 2*X* (1014L % − 50%), while the percentages of 1014F and NHEJ alleles on the receiver chromosome, also shown in red type within parentheses, were estimated by multiplying by a factor of 2). **e, f** Proportion of F$_2$ males or females inheriting the donor (*mW*) versus receiver (*w*[-]) chromosome. These F$_2$ progeny were collected from either F$_0$ grandfathers (**e**, *n* = 10) crosses or F$_0$ grandmothers (**f**, *n* = 13 crosses) carrying the e-Drive cassette. Data analyzed using a One-way ANOVA. Error bars indicate mean ± standard deviation, with the mean value indicated in the bar. Asterisks denote *p*-values: ****$p < 0.0001$, ns indicates not significant.

eliminated due to Cas9/gRNA cleavage activity at the *para*[1014F] target locus (Fig. 2e, f). This loss of receiver chromosomes in male F$_2$ progeny was not observed in our previous split allelic-drive studies (see section "Discussion" for potential factors contributing to this difference).

In F$_2$ female offspring, we observed no significant bias in transmission of donor versus receiver chromosomes (Fig. 2e, f), presumably reflecting the fact that females carry two X-chromosomes, one of which is intact (i.e., inherited from the *w1118* male F$_1$ parent) and can presumably complement lethal alleles generated by Cas9/gRNA cleavage of the target chromosome. We tested the hypothesis that a significant fraction of F$_2$ females carried recessive male lethal alleles by isolating and evaluating individual isogenic F$_2$ receiver chromosomes (by crossing heterozygous F$_2$ females to FM7 males to establish balanced lines carrying only single receiver chromosomes, Supplementary Fig. 3a). Indeed, ~24% of these balanced F$_3$ lines generated no male progeny carrying the isogenized receiver chromosomes and thus indicating the presence of male lethal alleles, likely resulting from NHEJ mutations induced at the *para*[1014F] target site (Supplementary Fig. 3b). We confirmed the presence of such lethal *para*[1014F] target alleles by sequencing the gRNA target site in female offspring of F$_3$ fly lines that produced only balancer carrying males. As predicted, females heterozygous for the balancer carried either frame-shift or mis-sense mutations at the target-site supporting our hypothesis (Supplementary Fig. 3c). In addition, we carried out Sanger sequencing of F$_3$ fly lines that did produce viable receiver-bearing male offspring to assess allelic 1014F->L conversion rates. Consistent with the allelic copying estimates summarized above, we observed that 20–28% of receiver *para*[1014F] alleles had been converted to *para*[1014L] (Supplementary Fig. 3b). These results reveal that the e-Drive element promotes transmission of the *para*[1014L] allele through two distinct and reinforcing mechanisms: 1) by allelic conversion from 1014F to 1014L, and 2) by target chromosome mutagenesis/elimination.

## The e-Drive efficiently reverses IR in multi-generational cages whether it disappears or persists in the population

We next tested the performance of the e-Drive in multi-generational cages. We conducted these experiments in two formats wherein the e-Drive either did, or did not, impose a strong male mating fitness cost. These two schemes rely on a classic example of sexual counter-selection, originally documented by Alfred F. Sturtevant[22], in which WT females strongly prefer to mate with WT *y*[+] over *y*[-] mutant males, with selection coefficients against the *y*[-] genotype ranging from −0.80 to −0.95[18–23]. Thus, in a *y*[+] genetic background, the e-Drive would be expected to suffer a severe reproductive disadvantage, as we have independently documented for genetic elements inserted into this locus[23,26–29]. In contrast, the e-Drive is expected to have a fitness-neutral phenotype in a *y*[-] background. We seeded parallel multi-generational cages in which the target *para*[1014F] allele was carried either in WT or *y*[-] genetic backgrounds

at two different ratios (1:3 and 1:1) relative to the target population. These experiments permit a comparison of performance and persistence of the drive element in contexts where the e-Drive incurs a strong male mating disadvantage (WT = *y*[+] background, or self-eliminating mode) versus one where it is on equal footing with the target population (*y*[-] mutant background, or hover mode) (Fig. 3a, b).

In self-eliminating mode, we seeded the e-Drive at either 25% (1:3) or 50% (1:1) (equal numbers of males and females), with the remaining 75% or 50% of flies, respectively, carrying the IR *para*[1014F] allele in a *y*[+] WT background (Fig. 3a, c, and e). In parallel experiments, conducted in hover mode, the target population carrying the *para*[1014F] allele was *y*[-] (Fig. 3b, d, and f). This second scheme allowed us to assess fitness costs associated with the e-Drive element per se and also to compare allelic-drive conversion performances when the cassette was present only transiently (*y*[+] target population), versus maintained in the population (*y*[-] target population).

At each generation, we scored half of the progeny in each cage for prevalence of the e-Drive element (based on presence of the DsRed[+] marker), and 25 randomly-selected flies from the same cages were processed to DNA extraction and deep sequencing at the *para*[1014F] locus (Figs. 3c–f, 4a–d). The other half of cage progeny were used to seed the next generation. We observed consistent similar initial e-Drive kinetics in all cages seeded in either self-eliminating or hovering mode. The percentage of DsRed[+] flies increased abruptly in generation 1, primarily as a result of random segregation of the dominantly marked (DsRed[+]) e-Drive bearing chromosomes (Figs. 3c–d, 4a–b) (note, however, that random chromosome assortment alone cannot fully account for the magnitude of this initial surge in e-Drive prevalence as discussed further below). In successive generations of cages seeded in self-eliminating mode (wherein the e-Drive element incurs a severe male mating fitness cost), we observed progressive elimination of the drive element over eight generations (1:3 seeding ratio) (Fig. 3c, Supplementary Fig. 4a, b) or after ten generations (1:1 ratio) (Fig. 4a, Supplementary Fig. 5a, b). Similarly, in experiments controlling for a potential fitness cost associated with the Cas9 transgene (e-Drive without vasa-Cas9), the e-Drive element was eliminated over 12 generations (1:1 ratio) (Supplementary Fig. 6a, b). In hover mode cages, without mating fitness costs imposed on the drive, the frequency of the DsRed marker remained stable with little variation across generations for both 1:3 and 1:1 seeding ratios, suggesting that no significant selective distortions were operative (Figs. 3d, 4b, Supplementary Figs. 4–6c, d).

In parallel to assessing prevalence of the e-Drive element, we evaluated the allelic frequencies of the *para*[1014L] versus *para*[1014F] alleles at each generation by NGS sequencing (Figs. 3e, f, 4c, d, Supplementary Figs. 4–6e–h). In cages seeded at a 1:3 ratio conducted in self-eliminating mode (with a strong fitness cost associated with e-Drive), we observed the WT *para*[1014L] allele increasing progressively from its 25% seeding frequency to ~80% by generations 8-9 despite the rapid

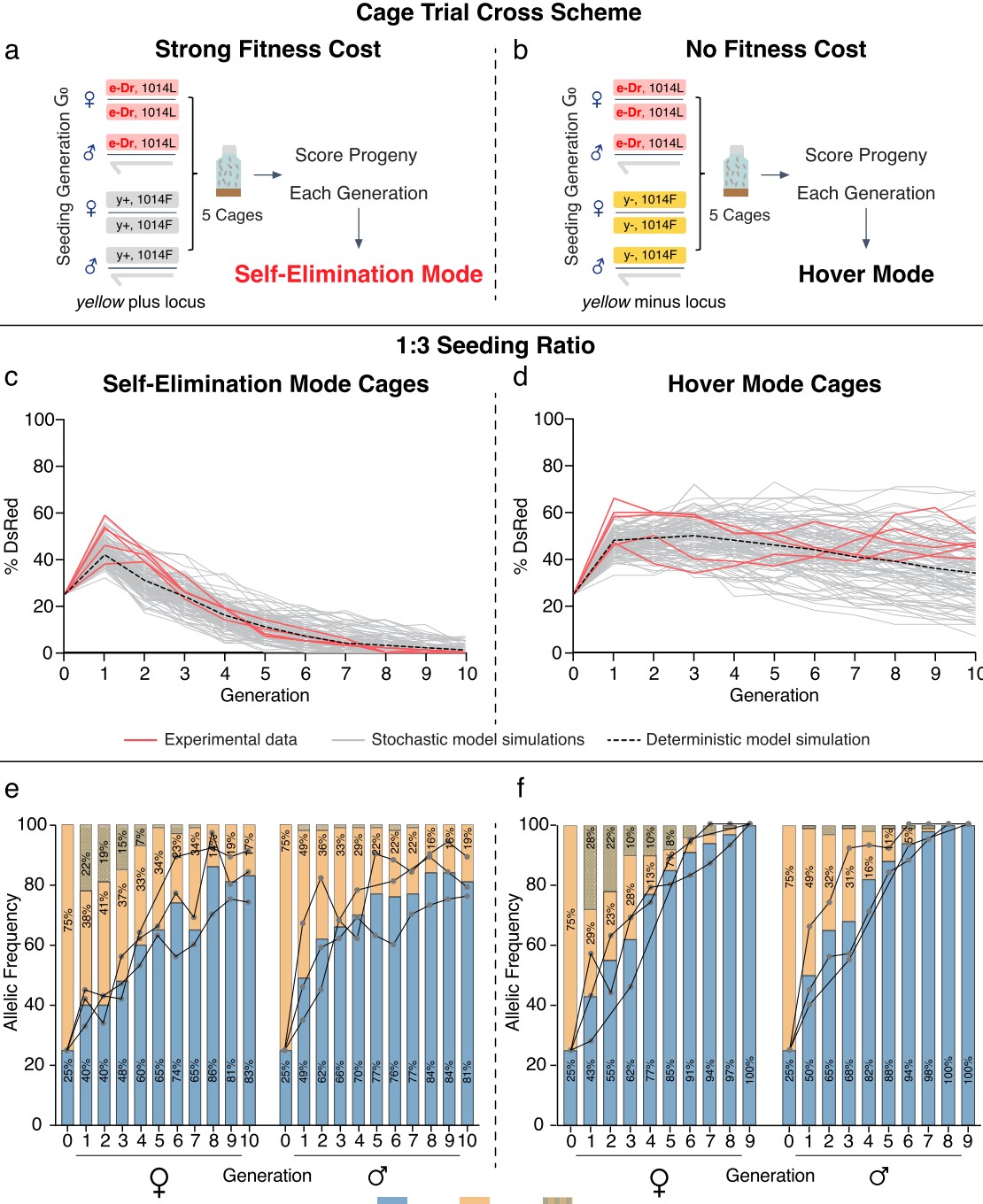

**Fig. 3 | The e-Drive reverses the IR phenotype in multi-generational cages seeded 1:3. a, b** Summary schemes for seeding multi-generational cages in self-eliminating mode (**a**), wherein the e-Drive (e-Dr) was introduced into a $y^+$,$para^{L1014F}$ background or in hover mode (**b**), wherein the e-Drive was introduced into a $y^-$,$para^{L1014F}$ background. **c, d** Percentage of individuals in population cages carrying the DsRed marked e-Drive as a function of generation. Initial e-Drive seeding frequency was 25%, with 75% seeding frequency of the $y^+$,$para^{L1014F}$ (**c**) or $y^-$,$para^{L1014F}$ target genotypes (**d**) ($N$ = 5 cages for both **c, d**). Colored traces represent experimental data (red = 5 independent replicates from two different experiments with 3 and 2 replicates), and deterministic (black) versus 100 stochastic (grey) model simulations. **e, f** Frequencies of different $para$ alleles (1014L; 1014F; NHEJ) sampled at each generation from self-elimination mode cages (**e**) or hover mode cages (**f**) ($n$ = 25 flies; $N$ = 3 cages for both **e** and **f**).

elimination of the drive element during the same time frame (Fig. 3e). These results are comparable to those observed previously with the split-drive system despite the persistence of the Cas9 source in the population throughout that prior experiment[16]. As expected, allelic conversion was even greater when the e-Drive was deployed in persistent hover mode where it incurred no fitness cost. In this case, where the e-Drive element remained stable in the population, conversion of the $para^{1014F}$ allele to $para^{1014L}$ eventually reached 100% (by generation nine—Fig. 3f). In control cages (e-Drive without vasa-Cas9) seeded at 50% frequency, the $para^{1014L}$ allele displayed a slower increase in frequency in both self-eliminating and hover modes (65.5% and 77% respectively by generation ten), (Supplementary Fig. 6e–h:—see also modeling extrapolations through generation 20, Supplementary Fig. 6i–l) presumably reflecting the previously documented negative

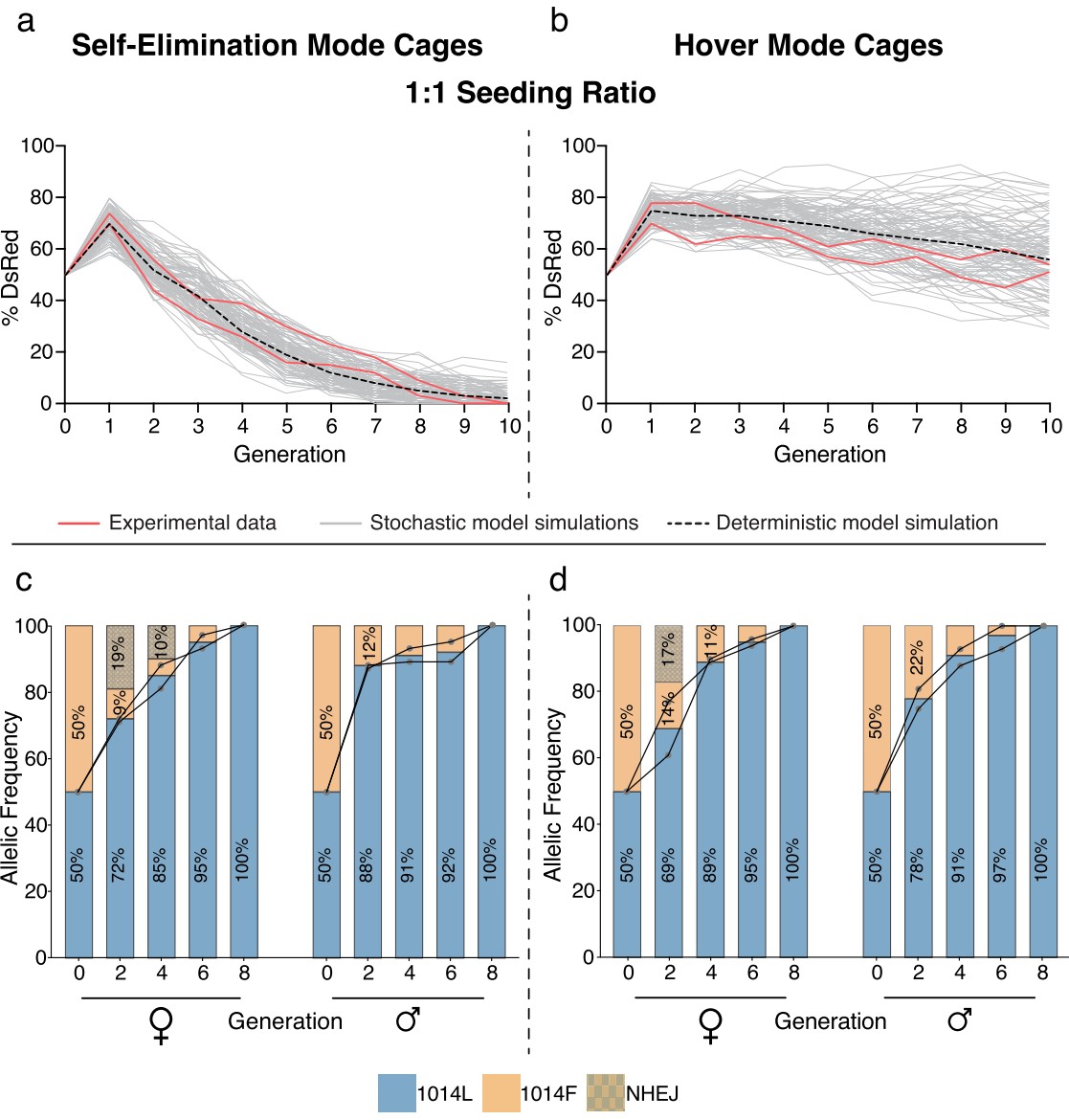

**Fig. 4 | The e-Drive fully reverses IR in multi-generational cages seeded 1:1.** Percentage of individuals carrying the e-Drive (DsRed⁺) at each generation. Initial e-Drive seeding frequencies was 50%, with 50% seeding frequency of *y+,para^L1014F* (**a**) or *y⁻,para^L1014F* (**b**) (*N* = 2 cages for both **a** and **b**). Red, grey, and dotted black curves represent, respectively, the experimental data (2 replicates), and deterministic or 100 stochastic model simulations. Frequencies of different *para* alleles (1014L; 1014F; NHEJ) sampled at each generation from self-elimination mode cages (**c**) or hover mode cages (**d**) (*n* = 25 flies; *N* = 2 cages for both **c** and **d**).

selection associated with the IR *para^1014F* allele relative to WT *para^1014F* [16] (see also Supplementary Fig. 7). Remarkably, when seeded at a 1:1 ratio, the e-Drive led to complete replacement of *para^1014F* by the *para^1014L* allele in either self-eliminating or hover modes, although the drive element was present for only an additional two generations in self-eliminating mode (Fig. 4c, d).

## Modeling reveals hidden aspects of cage drive dynamics

We developed a mathematical model to capture the dynamics of the e-Drive system and fitted it to data acquired from the multi-generational cage experiments using Markov chain Monte Carlo (MCMC) methods with no prior information included (Figs. 3c–d, 4a, b, Supplementary Figs. 4, 5 and 6). Dynamics of the e-Drive in the self-eliminating and hover modes were modeled considering discrete generations to account for the non-overlapping nature of the experiments. We employed a simplified framework in which we represented the main genotype features indicated in Supplementary Table 1. Because the *yellow* and *para* loci are located far apart on the X-

chromosome, ~50 cM, we treated their inheritance independently. We further assumed that the recombinant e-Drive system and *para^1014F* males were inviable since there would be no "template allele" to accurately repair the *para^1014F* locus in response to cleavage by the e-Drive system (an assumption verified by our experiments in which we recovered no viable NHEJ alleles in males). Trans-heterozygous NHEJ/ *para^1014F* and homozygous *para^1014F/para^1014F* females also carrying the e-Drive through recombination were similarly assumed to be inviable, again due to the lack of an intact *para^1014L* template available to repair the *para^1014F* allele upon cleavage by the e-Drive system. Full model details are included in the Supplementary Materials.

As noted above, the fraction of *para^1014L* alleles tallied in first cage generation cage experiments was greater than would be expected based on simple chromosome segregation, suggesting that a fitness cost was associated with the *para^1014F* allele, as we documented previously in *D. melanogaster* [16] and has also been reported in mosquitoes [30–32]. We confirmed this effect for the e-Drive by carrying out competitive mating experiments in which equal numbers of WT

*para^1014L* and *para^1014F* females were crossed to either to WT or e-Drive males (Supplementary Fig. 7). These crosses indeed revealed a nearly 2:1 bias in favor of transmission of the *para^1014L* allele (Supplementary Fig. 7). We therefore incorporated a corresponding fitness cost for males and females carrying the *para^1014F* versus *para^1014L* alleles into our model, which also aligns with the multi-generational data shown in Supplementary Fig. 6e-l.

The best fitting model suggested that the rate of e-Drive conversion from *para^1014F* to *para^1014L* is 27% (95% credible interval (CrI): 0–81%). Additionally, we estimated the rate of e-Drive conversion from *para^1014F* to *para*^lethal to be 32% (95% CrI: 0–84%). The model also predicted that males carrying the e-Drive system were 84% (95% CrI: 71–94%) less competitive in mating than those not carrying the e-Drive, and that in fitness-neutral hover mode, males carrying the e-Drive system were 26% (95% CrI: 8–40%) less competitive in mating than those not carrying the e-Drive. These estimates for the e-Drive in self-eliminating mode are consistent with a previously estimated bulk fitness parameter for a *y*^− gene cassette competing in a *y*^+ background[23].

With respect to *para^1014F* females, the best fitting model predicts that *para^1014F* homozygotes are 56% (95% CrI: 37–73%) less fertile than both *para^1014F* heterozygous or WT females, and furthermore that the increase of *para^1014L* frequencies throughout the generations can be partially explained by maternal Cas9 being transmitted to progeny carrying *para^1014F* (estimated to occur 47% of the time: 95% CrI: 2–93%). Overall, the modeling supports the conclusion that the rapid experimentally-observed drive dynamics (Supplementary Figs. 4, 5) results from a combination of lethality associated with either males or females inheriting a *para^1014F* allele together with the e-Drive, a substantial fitness cost associated with the *para^1014F* allele relative to the *para^1014L* allele, and super-Mendelian allelic conversion by the e-Drive element (full details of the model fitting are included in the Supplementary Materials).

## Discussion

The self-eliminating allelic-drive system described herein, carrying Cas9 and a single gRNA that targets the *para^1014F* allele, efficiently reverses the insecticide-resistant *para^1014F* genotype back to a WT *para^1014L* insecticide susceptible state when deployed in either self-eliminating or persisting hover modes. Consistent with our prior studies employing a two-component *para^1014F* reversal drive, the unitary e-Drive element proved highly efficient in driving the *para^1014L* allele into a *para^1014F* target population under fitness-neutral conditions. Throughout the duration of these experiments, the drive element maintained stable frequencies and reverted 100% of *para^1014F* alleles to the native *para^1014L* allele by generation nine when seeded even at low levels (1:3). Remarkably, the 1014F allele also was efficiently replaced by 1014L, albeit not as completely (~80%), when the e-Drive was deployed in self-eliminating mode and seeded at only a 1:3 ratio. Full allelic conversion was achieved in either self-eliminating or hover modes when the e-Drive was seeded at a 1:1 ratio.

A surprising element of these studies was that the unitary allelic-drive system, when deployed in self-eliminating mode, performed as well as the prior tested split allelic-drive (~80%) with regard to final levels of allelic conversion, even though the e-Drive was rapidly lost from the population in contrast to the separate Cas9 source, which remained at constant frequency in the population throughout the split-drive experiments[16]. Although we made a modest improvement to gRNA-F, the rates of allelic conversion with this optimized gRNA were similar to those observed in the original split system (20–40%) in F_2 tests. One important mechanism that modeling suggests is likely to contribute to the efficient allelic conversion in multi-generation cages with the e-Drive is its lethal effect when combined with the *para^1014F* allele either in hemizygous males or in *para^1014F*/*para^1014F*

homozygous females, which lack a protected and functional *para^1014L* repair template. Such situations would most likely lead to a strong form of lethal mosaicism[17,33,34], in which *para^1014F* alleles were efficiently mutated to non-functional NHEJ alleles resulting in loss of *para* function in a large fraction of the nervous system. Also, the enhanced performance of the unitary e-Drive relative to the split-drive may derive partly from the gRNA and Cas9 always being co-inherited as a unit for the e-Drive, while in the case of the split-drive, the gRNA and Cas9 often separate due to random assortment of the Cas9 and gRNA bearing elements. Persistent multi-generational linkage of the Cas9 and gRNA may lead to progressive accumulation of higher levels of maternally-transmitted Cas9/gRNA ribonucleoprotein complexes that more efficiently cleave and/or convert the targeted *para^1014F* allele. Indeed, we observed a similar elevation in apparent maternally-based cleavage activity when comparing full-drive[35] to split-drive[33] systems inserted into the *spo11* locus. Another important contribution to the success of the e-Drive (as well as for the prior *kdr* split drive) are the substantial recessive fitness costs associated with the *para^1014F* allele, which include temperature-sensitive reduced mobility and sleep defects when compared to the WT *para^1014L* allele[16] (Supplementary Fig. 7). Finally, it is also possible that alternative DNA repair pathways may be differentially recruited to repair DSBs in different contexts. For example, in contrast to HDR-mediated copying of gene cassettes, which require a step of DNA synthesis, repair of DSBs only require repair of mismatches encompassing a few nucleotides between the donor and receiver chromosomes. Such localized repair could potentially involve recruitment of processes including Mismatch Repair, Base Excision Repair, Nucleotide Excision Repair, or Translesion Synthesis.

The e-Drive design addresses one of the most pressing issues in the gene-drive engineering community[36]. Because this gene cassette does not carry a gRNA element to copy itself, it either disappears rapidly from the population (when incurring a fitness cost) or persists at a constant frequency (when deployed in a fitness-neutral hover mode). In self-eliminating mode, during the brief time the e-Drive is present in the population (that duration being adjustable based on its introduction frequency), gRNA-F can convert the non-preferred IR allele back to WT, culminating in a non-GMO endpoint wherein the only final effect is an alteration in the ratio of IR versus WT *vgsc* alleles in the population. These findings provide experimental support for employing the e-Drive in potential field studies and should contribute to gaining community and regulatory approval for such releases. Flexible deployment of the e-Drive at differing initial seeding frequencies in self-eliminating or hover modes would be expected to achieve efficient IR reversal under diverse conditions and should be generalizable to other loci such as the FREP1 gene in anopheline mosquitos to bias the inheritance of a functional allelic variant that reduces transmission of malarial parasites[37]. In cases where target loci for allelic modification are either essential for viability (e.g., *vgsc*) or fertility (e.g., *FREP1*), one could generate equivalent systems acting either in hover or self-eliminating modes by inserting the editing cassette into conserved protein-coding sequences and either include (hover mode), or not (self-eliminating mode), 3' recoded sequences to maintain target gene function[33,34]. In case of anopheline mosquitos where the *vgsc or FREP1* genes are located autosomally, the observed e-Drive dynamics may differ from *Drosophila*. There are two notable differences expected for an autosomal *vgsc* target allele. Firstly, loss-of-function *vgsc* alleles generated by the action of the e-Drive would not be hemizygous lethal in males of the next generation. The removal of such lethal alleles would thus be delayed compared to the X-linked situation, where 50% of lethal *vgsc* alleles are immediately removed from the population. Secondly, the allelic-drive should take place in both male and female parents for an autosomal *vgsc* target, which

should speed up the drive process. The balance between these two opposing effects is likely to depend on various parameters such as the rate of generating NHEJ alleles and the copying efficiency of the e-Drive element. In addition, an e-Drive could be equipped either with genetically encoded RNAi constructs or with truncated guide RNAs (tgCRISPRi/a) enabling scarless transcriptional modulation to repress expression of NHEJ pathway components or to activate HDR pathway genes, as reported previously[38,39], in efforts to increase allelic copying efficiency via HDR.

An obvious constraint applying to deployment of genetic systems for reverting IR in real-world contexts is that the relevant insecticide should not be in use at the time the genetic reversal system is being deployed. One way this could be achieved would be to alternate between two different insecticides, using the insecticide that acts on the non-targeted locus during genetic reversion treatment. This strategy would require developing two self-eliminating systems to revert resistance to each compound (see ref. 16 for a more in-depth discussion of this issue and inclusion of alternative IR targets). In the case of *kdr* reversion strategies, it might also be possible to take advantage of cases in which mosquitoes feed locally on both livestock and humans[40] by limiting the use of pyrethroids to bed nets or indoor sprays and then apply the IR reversal drive to areas of zoophilic biting such as livestock housed in adjacent pens or fields. If such zoophilic populations of mosquitoes bred and inter-mated at some significant frequency with those biting humans indoors, perhaps the percentage of IR mosquitoes in the overall local population could be reduced sufficiently to render the insecticide interventions more effective. Indeed, modeling of bed net efficacy relative to IR frequency in one study suggest that they could still provide substantial protection when IR frequencies are as high as 50–60%[41]. While a spatially segregated replacement strategy would result in substantially slower kinetics for IR reduction than deploying the e-Drive in the complete absence of target insecticide use, such a procedure could be repeated to maintain a countervailing selective pressure opposing the fitness advantage of individuals with an IR phenotype.

It may also be possible to leverage fitness costs as primary drivers for IR reversion of populations. For example, since IR alleles such as *kdr* mutations often impose appreciable fitness costs in field as well as laboratory contexts[42,43] (although such costs can be variable and subject to suppression by second-site modifiers[44]), one might simply follow a strategy of repeated inundative releases of insects carrying the wild-type insecticide-sensitive allele during periods when the target insecticide is not in use. Alternatively, one might take advantage of the negative selection imposed by insecticides to generate a self-eliminating behavior of a unitary drive cassette by inserting it within a non-essential site of the IR target locus (e.g., in an intron), and linking it to the WT protected target allele. The gene cassette should then disappear following the reapplication of the target insecticide.

In summary, our study strongly supports self-eliminating allelic-drive strategies as a broadly applicable means to achieve population replacement of an undesired allele with a different naturally occurring preferred allele in diverse experimental, agricultural, and natural contexts. Specifically, these proof-of-principle experiments demonstrate the feasibility of reverting target site IR alleles such as that conferred by *para^1014F* to the native *para^1014L* insecticide susceptible state with a zero GMO endpoint, an attractive initial system to consider for candidate field release trials.

## Methods

### Fly strains

*y^<CC|pF|>* allelic-drive and *para^1014F* fly strains were generated in previous study by Kaduskar et al., vasa-Cas9-GFP (BL# 79006) and BL# 15811 strains were procured from Bloomington *Drosophila* Stock Center (BDSC). e-Drive line used in the study were generated in-house.

### Construction of e-Drive element and transgenic flies

We followed the cloning strategy described in Guichard et al., wherein homology arms abutting gRNA-y1 cleavage site of the *yellow* locus and carrying gRNA-F, vasa-Cas9, and a 3XP3-DsRed eye marker were assembled in the plasmid as shown in Fig. 1. We made slight modifications to the gRNA-F, which initially had two guanine nucleotides at the 5′ end of 20 nt gRNA-F, by replacing them with a single guanine nucleotide at the 5′ end of gRNA-F. We amplified sequences using Q5 Hotstart Master Mix 2x (New England BioLabs, Cat. #M0494S) and PCR fragments were assembled with NEBuilder HiFi DNA Assembly Master Mix (New England Biolabs, Cat. #E2621). The resulting plasmids were transformed into chemically competent 5-alpha *E. coli* (New England BioLabs, Cat. #C2987), isolated, and sequenced. The complete sequence of the assembled plasmid was verified and deposited at GenBank accession number PP695280.

The e-Drive and helper pCFD3-gRNA-y1 plasmids were injected into *w1118* stock by Rainbow Transgenics. In $F_1$ progeny, we identified male transformants carrying the e-Drive element by their *yellow^-* and DsRed^+ eye fluorescence, followed by sequence validation of the inserted cassette of PCR amplified genomic sequences.

### Sample preparation for sequencing

For each experiment replicate, a group of randomly selected 20–25 male or female flies was crushed in 500 µl of homogenization medium consisting of 0.1 M Tris, 0.1 M EDTA, 1% SDS, and 0.5% diethylpyrocarbonate. The mixture was then incubated at 65 °C for 30 min. The DNA was extracted by precipitating it with 100 µl of 8 M potassium acetate and isopropanol[45]. Subsequently, the gDNA target sequence was amplified using vgsc-specific primers with adapter sequence and finally run on Illumina NOVOseq platform. vgsc forward: 5′ ACACTCTTTCCCTACACGACGCTCTTCCGATCT agcttcatgatcgtgttcc 3′. vgsc reverse: 5′GACTGGAGTTCAGACGTGTGCTCTTCCGATCT gccatggttagaggcgataagtc3′. The data obtained were analyzed using CRISPResso2 pipelines[46].

### Fly genetics and crosses

*e-Drive*, m*W*, *para^1014L* and *y^+*, *w^-*, *para^1014F* fly stocks were maintained on regular cornmeal medium under standard conditions at 18 °C with a 12-h day-night cycle. Crosses were performed in glass vials in an ACL1 fly room, freezing the flies for 48 h before discarding. To assess each allelic-drive efficiency, we crossed either male or female flies carrying the e-Drive construct to the opposite sex carrying the *para^1014F* allele. Since our gene of interest is on X-chromosome, individual trans-heterozygote $F_1$ virgin females were collected for each $G_0$ cross and crossed to a wild-type fly of the opposite sex. Single one-on-one crosses were grown at 25 °C. We calculated donor and receiver chromosome inheritance using the resulting $F_2$ progeny by scoring the presence or absence of *mW* red eye phenotypic markers, respectively. We analyzed the receiver allelic conversion frequency by NGS sequencing of the *para* locus of the *w^-* receiver $F_2$ male and female progeny. In *w^-* $F_2$ females, which inherited one X-chromosome from $F_1$ male parent allele and other receiver X-chromosome from $F_1$ female parent allele, the rate of receiver allelic conversion was calculated using the $2x(Y - 50\%)$ formula. Y represents the total observed frequency of L reads, with the deduction of 50% based on the assumption that half of the reads sequenced by NGS come from the reference $F_1$ male parent allele.

### Experimental setup for cage studies

All cage experiments were conducted at 25 °C with a 12-h day-night cycle. Glass bottles containing standard cornmeal medium were used for the experiments. The cages were seeded at a ratios of either 1:3 or 1:1. For cages seeded at 1:3 ration, 25% of flies carrying *e-Dr, para^1014L* (15 males and 15 females) and 75% of the flies carrying *y^+,w^-, para^1014F* or

$y^-$, $w^-$, $para^{1014F}$ (45 males and 45 females). In cages seeded at a 1:1 ratio, 50% of flies carrying $e$-$Dr$, $para^{1014L}$ (30 males and 30 females) were combined with 50% of flies carrying $y^+$, $w^-$, $para^{1014F}$ or $y^-$, $w^-$, $para^{1014F}$ (30 males and 30 females). In each generation, the flies were allowed to mate and lay eggs for approximately 72 h. The parents ($G_0$) were then removed, and cages were kept for 10 days. The subsequent progeny ($G_1$) were randomly separated into two pools. One pool was collected for scoring and sequencing analysis, while the other was used to seed the next generation. This process of sampling and passage was continued for up to 20 generations.

## Statistics and reproducibility

This study included all offspring that were produced from the given cross and population cages. However, crosses with fewer than 10 offspring were not included, as the low number could have been caused by contaminated food. We also randomly-selected flies for Sanger sequencing analysis and NGS analysis. No data was excluded in this study. We performed multiple replicates of the crosses and cage setups, resulting in the generation of over 10 offspring per cross and over 100 offspring per cage. The replicates yielded consistent and comparable results, indicating that the experimental conditions were reliable and reproducible. Progeny were randomly selected from crosses and cages.

## Reporting summary

Further information on research design is available in the Nature Portfolio Reporting Summary linked to this article.

## Data availability

The e-Drive plasmid sequence data have been deposited in the NCBI database under accession code PP695280. All data can be found in the main text or the Supplementary materials. The raw data and model fitting data generated in this study are provided in the Source Data file. Source data are provided with this paper.

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

## Acknowledgements

We thank all members of the Bier laboratory and the anonymous reviewers for constructive ideas and discussions and funding sources: The Bill & Melinda Gates Foundation INV-036579 (E.B.), The Bill & Melinda Gates Foundation INV-017683 (J.M.), National Institutes of Health grant R01GM117321 (E.B.), National Institutes of Health grant R01GM144608 (E.B.), National Institutes of Health grant R01AI162911 (E.B.), and the Tata Institutes of Genetics and Society, UCSD (E.B.).

## Author contributions

A.A.: Designed and contributed to conducting experiments and writing the original draft. J.M.M. and R.M.C.: Contributed to modeling experiments and editing the manuscript. E.B.: Contributed to the original concept of the e-Drive and writing the original draft.

## Competing interests

E.B. has equity interest in Synbal, Inc. and Agragene, Inc., companies that may potentially benefit from the research results. The terms of this arrangement have been reviewed and approved by the University of California, San Diego in accordance with its conflict-of-interest policies. The other authors declare no competing interests.
