## [Transparent Peer Review file · Nature Communications]

A self-eliminating allelic-drive reverses insecticide resistance in *Drosophila* leaving no transgene in the population

Corresponding Author: Professor Ethan Bier

Version 0:

Reviewer comments:

Reviewer #1

(Remarks to the Author)

This is an elegant and thorough development of the earlier work by the same authors on a split drive cassette system for replacing mutants conferring insecticide resistance with the original wild type allele. The work represents a major step forward combining the elements of the split drive system into a single eDrive cassette which would lend itself more easily to deployment in field settings.

A major advantage of the new approach is the clearance of the genetically manipulated material with a relatively small number of generations, which should remove many of the objections to release of self propagating genetically manipulated material in the germlines of wild populations.

The authors assess the effects of different phenomena, such as the non-homologous end joining potential issues creating increased target site mutations during early embryogenesis in the female germline. Using a series of appropriate experiments they are able to account for the loss of receiver X chromosomes in the males by the F2 generation of their cage experiments which occurs in the eDrive but not in the split cassette system.

In discussing the potential use of this in field settings they suggest that they system would need to be deployed while the wild population was not under active selection by the insecticide(s) to which the resistance gene confers a selective advantage. Their cage and modelling work have also been done in the total absence of any insecticide selection pressure. While this is appropriate at this stage of the developmental cycle of the technology it is important that the impact of any positive selection by the IR is factored in as the technology is developed. While it is possible to control the insecticides that are specifically sprayed against *Anopheles* in indoor residual spraying or on the bednets that are distributed, alongside this the same insecticides will be used in the same locations for agricultural purposes. The trend for long lasting insecticide treated bednets is also rapidly switching to dual treated nets - the first generation having a mixture of PBO + pyrethroid - which effectively try to block out the multi-function oxidase pyrethroid resistance and the newer nets with two active insecticides, one often being a pyrethroid. As the modelling is progressed it would be good to see how the eDrive technology might prolong the active life of these dual acting nets by retaining the efficacy of the pyrethroids and reducing the potential for resistance selection to the second insecticide and I would like to see these flagged in the discussion.

Reviewer #2

(Remarks to the Author)

This work builds on previous work targeting the allelic conversion of an insecticide-resistant allele of a voltage-gated sodium channel gene called paralytic (*para*) in *Drosophila melanogaster*. *Para* is an essential gene located on the X chromosome and (non-functional) mutations in this gene are recessive lethal in females and lethal in males (hemizygous). The insecticide resistant (IR) allele (1014F) is associated with several fitness costs while the wild type allele (L1014) is the preferred allele in the absence of selection pressure imposed by pyrethroid insecticides such as DDT.

This allelic drive benefits massively from the wild type insecticide-susceptible (IS) allele (L1014) of *para* being adaptively preferred over the mutant version. This is because the mutant version is recessive lethal and is associated with several

fitness costs. This feature of the target site enables the non-homing Cas9 located in and disrupting the yellow locus (disruption results in severe male mating fitness cost) to successfully convert the IR alleles to IS alleles within a brief span of ~ 10 (1:3 seeding ratio) or 8 (1:1 seeding ratio) generations in both self-elimination and hover modes. It is interesting that at 1:1 seeding ratio, the mode doesn't matter when it comes to the number of generations needed to convert to and fix the IS wild type allele in the discrete populations. This makes the multigenerational control crosses interesting as the strong positive selection for the wild type IS allele increases its frequency from a starting point of 50% to ~70% in 10 generations (Sup.Fig.6e-h). What would be the allele frequency of IS alleles in such a population without any drive system, in about 15 or 20 generations, in the absence of pyrethroid insecticides? Would the IS allele reach near fixation even without what seems to be a minor boost (allelic conversion rates of 7-40%) that the e-Drive provides? If so, wouldn't high threshold releases of individuals harboring wild type preferred IS alleles for one class of insecticides, say P (while a different class of insecticides, say O, are in operation) be able to make the population susceptible to the P class of insecticides in 15-20 generations? The same suggested idea of alternating pesticide classes albeit with high-threshold release of non-transgenic non-drive individuals with insecticide class-specific IS alleles would still apply, however, without any further regulatory approvals being needed, as both the use of pesticides and release of GM mosquitoes are currently approved. Just some food for thought as this high-threshold approach of releasing preferred wild type alleles (PIT: Preferred allele Insect Technique or perhaps PAT = Preferred Allele Technique) could be deployed right now. If pesticide use is unavoidable, might as well start now with high-threshold releases of non-transgenic preferred alleles under strong positive selection, no?

Having said the above, where pesticides aren't involved, the allelic e-Drive presented here may vastly benefit the introduction of pathogen-refractory alleles that are "adaptively preferred." Many of the gene drives, designed and tested so far, carry fitness costs relative to the wild type organism, which is expected when disrupting a gene (especially an essential gene) or introducing transgenic constructs (the long-term effects of which, especially in the wild, cannot be accurately predicted). So, as suggested, whether in self-eliminating mode (high fitness costs associated with the transgene disrupting an essential gene or a gene introducing mating or metabolic fitness costs) or hover mode (recoded transgenes disrupting essential genes or mutant background in this case), a preferred allele that confers the vector refractoriness to a pathogen could be successfully driven to fixation along with the elimination of the pathogen.

This is a well-structured and excellent piece of work, my comments are as follows:

1. Lines 98-99:

80% alleles of para1014F being mutated makes sense to me if I double the 40% NHEJ alleles (considering only 50% alleles are mutable to begin with) shown in Sup.Fig.1a. However, this may not be intuitive to a novice reader generally interested in gene drive technology, perhaps a clarification of some sort would help. This clarification may help this novice reader with lines 117 through 120. Please note that this is a nitpick for the sake of the novice reader.

2. Lines 136-137:

The loss of receiver chromosomes in the e-Drive configuration vs the split drive configuration could be influenced by the chromosome Cas9 is encoded on. In the split-drive configuration, the Cas9 was encoded on an autosome whereas the e-Drive encodes it on the X-chromosome. Differences in replication and expression patterns when encoded on sex chromosomes vs autosomes may contribute to higher para-associated non-functional allele-driven lethal mosaicism, that may explain the substantive loss of receiver chromosomes in the e-Drive. Thought experiment, if not a real one: If the Cas9 e-Drive part were encoded on say the ebony locus on the 3rd chromosome, would we still see the same substantive loss of receiver chromosomes, due to lethal mosaicism, at the para locus on the X-chromosome?? Something to ponder upon. I refer to the sex vs autosome debate as I have seen some strange things happen to transgenes (specifically Cas9) when encoded on the X chromosome vs an autosome. Perhaps, future designs should shift to target genes on the autosomes or may include both as a comparison!! Perhaps include the X vs Autosome differences in the discussion, if feasible.

3. Lines 147-157:

The grandmaternal lineage is generally thought to be associated with higher rates of NHEJ alleles, likely due to cumulative generational maternal Cas9 deposition. It is interesting that Fig.2c & d show relatively higher F->L conversion and lower NHEJ rates in the maternal lineage (F0 Female) vs the F0 male lineage. Could this difference be due to the repair in this case being carried out by HR (DSB) in association with MMR/BER/NER or even TLS followed by MMR/BER/NER. This is possible as the target site at para needs to only repair a mismatch of 2 nucleotides between the donor and receiver chromosomes. HR of the DSB will result in two mismatches, which might be repaired by recruitment of one or more of the aforementioned repair mechanisms.

MMR = MisMatch Repair

BER = Base Excision Repair

NER = Nucleotide Excision Repair

TLS = TransLesion Synthesis

Some thoughts on this would be helpful given the widespread data now available on the so called grandmaternal effect or GESP and will also help expand the discussion around the recruitment of other DNA repair pathways that are likely to be recruited during DSB repair that act in concert with HR. Such a discussion might also strengthen the case of allelic-drive at a distance (reminds me of Einstein's description of quantum entanglement as 'spooky action at a distance') where perhaps DNA repair mechanisms other than HR are recruited leading to higher F->L conversion rates. I think that this discussion is

missing from the gene drive literature and should be shed light on.

Additionally, in Fig.2d, F2 receiver males, that uncover the allelic conversion rates in the maternal germline, are at 7% (F0 male) and 13% (F0 female) vs the allelic conversion rates in receiver females (Fig.2c) where almost 4 times higher rates are observed (28% and 40%). This is likely the result of both maternal germline conversion as well as zygotic conversion (possibly via MMR, TLS and other repair mechanisms). One could clarify this point.

A different question that arises is in the FM7 crosses intended to uncover lethal alleles, here maternal germline conversion rates of F to L are at 28% (F0 male) and 20% (F0 female). This is 4x or ~1x higher than in Fig.2d. Is it possible that the para target site in FM7 is accessible to targeting by the e-Drive and subsequent sequencing reveals a higher rate of F to L conversion? An inversion on the balancer chromosome may not curtail the Cas9 RNP from targeting a short 20 bp sequence, no?

Lines 147-149, about the ~24% balanced F3 lines not producing any viable males carrying the receiver chromosome, the figure Sup.Fig.3 is a bit confusing to me. If I understand correctly, the 22 and 24% male alleles are actually a % of the number of pair-mated crosses that did not generate balancer-free or receiver chromosome-carrying male progeny. This cannot be sequence-based as one cannot sequence the inviable. If so, how were the 1014L and F alleles frequencies recorded? Are these derived from sequencing the F3 fly lines mentioned in lines 154-157? Were no functional resistant alleles recorded? I am super curious. I think what confused me is the combination of recorded number of crosses with sequencing-based allelic frequencies. Please clarify if I am getting something wrong here as I am a bit confused.

Lines 150-151: "female offspring of F3 fly lines." Is my understanding that, female progeny from the F3 balanced lines established and maintained as stocks (progeny of F3 and future generations) were sequenced, accurate?

Line 156: Should it be 20-28% to match Sup.Fig.3b?

Reviewer #3

(Remarks to the Author)

Version 1:

Reviewer comments:

Reviewer #1

(Remarks to the Author)

The authors have responded appropriately to all the comments from both myself and the other reviewers

Reviewer #2

(Remarks to the Author)

The authors have addressed all of my comments in their rebuttal. The only possible omission is the X chromosome vs autosome issue that although they said they have addressed in the discussion of the paper, I couldn't find this in the discussion.

Reviewer #3

(Remarks to the Author)

REVIEWER COMMENTS

Reviewer #1 (Remarks to the Author):

This is an elegant and thorough development of the earlier work by the same authors on a split drive cassette system for replacing mutants conferring insecticide resistance with the original wild type allele. The work represents a major step forward combining the elements of the split drive system into a single eDrive cassette which would lend itself more easily to deployment in field settings.

A major advantage of the new approach is the clearance of the genetically manipulated material with a relatively small number of generations, which should remove many of the objections to release of self propagating genetically manipulated material in the germlines of wild populations.

The authors assess the effects of different phenomena, such as the non-homologous end joining potential issues creating increased target site mutations during early embryogenesis in the female germline. Using a series of appropriate experiments they are able to account for the loss of receiver X chromosomes in the males by the F2 generation of their cage experiments which occurs in the eDrive but not in the split cassette system.

In discussing the potential use of this in field settings they suggest that they system would need to be deployed while the wild population was not under active selection by the insecticide(s) to which the resistance gene confers a selective advantage. Their cage and modelling work have also been done in the total absence of any insecticide selection pressure. While this is appropriate at this stage of the developmental cycle of the technology it is important that the impact of any positive selection by the IR is factored in as the technology is developed. While it is possible to control the insecticides that are specifically sprayed against Anopheles in indoor residual spraying or on the bednets that are distributed, alongside this the same insecticides will be used in the same locations for agricultural purposes. The trend for long lasting insecticide treated bednets is also rapidly switching to dual treated nets - the first generation having a mixture of PBO + pyrethroid - which effectively try to block out the multi-function oxidase pyrethroid resistance and the newer nets with two active insecticides, one often being a pyrethroid. As the modelling is progressed it would be good to see how the eDrive technology might prolong the active life of these dual acting nets by retaining the efficacy of the pyrethroids and reducing the potential for resistance selection to the second insecticide and I would like to see these flagged in the discussion.

This is an excellent point and a key question to consider when thinking of implementing an effort to revert IS using a self-eliminating system in a real world context. As the reviewer notes, all current insecticide formulations for bed nets include pyrethroids. One possible approach to deploying an eDrive-like element in the context of bed nets containing a pyrethroid mixture would be to target nearby populations of insects that feed on non-human animals such as livestock housed in adjacent pens or fields. In some

settings there is a substantial fraction of zoophilic blood feeding (e.g., as reviewed in Barreaux et al., 2017, *Trends in Parasitology* **33**, 763-74). If these zoophilic populations of mosquitoes bred in overlapping locations and inter-mated at some significant frequency with those biting humans indoors (where pyrethroids would perhaps be used both in bednets and for indoor residual spraying) then perhaps the percentage of IR mosquitoes in the overall local population could drop sufficiently to render the insecticide interventions more effective. Indeed, modeling of bed net efficacy relative to IR frequency in one study (Churcher et al., 2016, *eLife* **5**, e16090) suggested that bednets could still provide substantial protection when IR frequencies are as high as 50-60%. While a spatially segregated replacement strategy would result in a substantially slower process for IR reduction than deploying the eDrive in the complete absence of target insecticide use, such a procedure could be repeated to maintain a countervailing selective pressure opposing the fitness advantage of individuals with an IR phenotype.

As the reviewer points out, a potential alternative would be to develop an eDrive that reverses resistance to a second insecticide or an additive such as PBO used in combination with pyrethroids. Such gene targets could include those providing metabolic resistance. These eDrives potentially could act by either deleting duplicate copies of detoxifying enzymes or by replacing mutant overactive promoter sequences with wild-type properly regulated control regions. We have added a short section to the revised discussion considering these alternative possible strategies.

Reviewer #2 (Remarks to the Author):

This work builds on previous work targeting the allelic conversion of an insecticide-resistant allele of a voltage-gated sodium channel gene called paralytic (*para*) in *Drosophila melanogaster*. *Para* is an essential gene located on the X chromosome and (non-functional) mutations in this gene are recessive lethal in females and lethal in males (hemizygous). The insecticide resistant (IR) allele (1014F) is associated with several fitness costs while the wild type allele (L1014) is the preferred allele in the absence of selection pressure imposed by pyrethroid insecticides such as DDT.

This allelic drive benefits massively from the wild type insecticide-susceptible (IS) allele (L1014) of *para* being adaptively preferred over the mutant version. This is because the mutant version is recessive lethal and is associated with several fitness costs. This feature of the target site enables the non-homing Cas9 located in and disrupting the yellow locus (disruption results in severe male mating fitness cost) to successfully convert the IR alleles to IS alleles within a brief span of ~ 10 (1:3 seeding ratio) or 8 (1:1 seeding ratio) generations in both self-elimination and hover modes. It is interesting that at 1:1 seeding ratio, the mode doesn't matter when it comes to the number of generations needed to convert to and fix the IS wild type allele in the discrete populations. This makes the multigenerational control crosses interesting as the strong positive selection for the wild type IS allele increases its frequency from a starting point of 50% to ~70% in 10 generations (Sup.Fig.6e-h). What would be the allele frequency of IS alleles in such a population without any drive system, in about 15 or 20 generations, in the absence of pyrethroid insecticides? Would the IS allele reach near fixation even without what seems

to be a minor boost (allelic conversion rates of 7-40%) that the e-Drive provides? If so, wouldn't high threshold releases of individuals harboring wild type preferred IS alleles for one class of insecticides, say P (while a different class of insecticides, say O, are in operation) be able to make the population susceptible to the P class of insecticides in 15-20 generations? The same suggested idea of alternating pesticide classes albeit with high-threshold release of non-transgenic non-drive individuals with insecticide class-specific IS alleles would still apply, however, without any further regulatory approvals being needed, as both the use of pesticides and release of GM mosquitoes are currently approved. Just some food for thought as this high-threshold approach of releasing preferred wild type alleles (PIT: Preferred allele Insect Technique or perhaps PAT = Preferred Allele Technique) could be deployed right now. If pesticide use is unavoidable, might as well start now with high-threshold releases of non-transgenic preferred alleles under strong positive selection, no?

The reviewer raises a very interesting potential strategy wherein one might just release wild-type IS insects in during periods when the insecticide is not being used and then rely only on the relative fitness cost of IR versus IS phenotypes to drive the IS genotype into the population. We carried out modeling simulations extrapolating the time courses of the experimental control cages (no Cas9) through 20 generations (See newly added panels: Suppl. Fig. 6i-l). The IS allele averaged 80-90% (dark blue curves) in both self-elimination and hover mode (attached figure A-D) by generation 20. In many stochastic simulations (light blue curves) the IR allele was completely eliminated from the cages.

However, there are two factors to consider here. First, although there is certainly a very significant fitness cost associated with the *kdr* allele in our laboratory setting (note the 1014F allele is temperature sensitive and carries a variety of other significant fitness costs, but it is not lethal at room temperature, while all NHEJ mutations generated by the gRNA-1014F are lethal). It is not clear, however, that such a strong fitness cost is associated with common 1014F/H/S *kdr* alleles in the field. For example, several studies have found that IR alleles can persist for several years in the absence of spraying a particular insecticide. Similarly, although the cyclodiene insecticide dieldrin has not been in use for over 20 years in some areas, the *rdl* allele of its target, the GABA-gated chloride channel, which confers resistance to dieldrin has persisted at high frequency in such long untreated mosquito populations. Fitness costs for *kdr* can also vary substantially between species. Thus, while particular *kdr* alleles such as the 1014F mutation have been

recovered in a broad range of insects, and many times independently in some species such as *Anopheles gambiae* across Africa, this mutation has not been found in wild populations of *Drosophila*. One possibility explanation for the absence of 1014F alleles in wild *Drosophila* is that the fitness costs are much greater in this species (temperature sensitive semi-lethality) than in other insects potentially due to differing destabilizing effects that this mutation has on the VGSC protein. Also, second site mutations can arise either within the VGSC gene or at second sites that greatly reduce the fitness costs associated with the primary 1014F allele. That said, in such cases where a *kdr* allele did not impose strong fitness costs, one could still envision inundative release strategies wherein an excess of IS insects were released to dilute the IR genotype. As discussed above in response to the point raised by Reviewer 1, this type of approach might be most productive for reducing *kdr* frequencies in mosquitoes if such IS releases were targeted to areas of zoophilic biting adjacent to those where human biting took place. We have added a short section to the discussion raising the possibility of employing this simple non-GMO approach.

Having said the above, where pesticides aren't involved, the allelic e-Drive presented here may vastly benefit the introduction of pathogen-refractory alleles that are "adaptively preferred." Many of the gene drives, designed and tested so far, carry fitness costs relative to the wild type organism, which is expected when disrupting a gene (especially an essential gene) or introducing transgenic constructs (the long-term effects of which, especially in the wild, cannot be accurately predicted). So, as suggested, whether in self-eliminating mode (high fitness costs associated with the transgene disrupting an essential gene or a gene introducing mating or metabolic fitness costs) or hover mode (recoded transgenes disrupting essential genes or mutant background in this case), a preferred allele that confers the vector refractoriness to a pathogen could be successfully driven to fixation along with the elimination of the pathogen.

We fully agree with the reviewer on this point.

This is a well-structured and excellent piece of work, my comments are as follows:

1. Lines 98-99: 80% alleles of para1014F being mutated makes sense to me if I double the 40% NHEJ alleles (considering only 50% alleles are mutable to begin with) shown in Sup.Fig.1a. However, this may not be intuitive to a novice reader generally interested in gene drive technology, perhaps a clarification of some sort would help. This clarification may help this novice reader with lines 117 through 120. Please note that this is a nitpick for the sake of the novice reader.

We appreciate this point and have added clarifying additional text to the legends of both Sup.Fig.1a and Fig. 2c for how we estimated the conversion frequencies in females according to the formulas: $2X$ NHEJ% for Supl.Fig.1a, and $2X (1014L\% - 50\%)$ for Fig. 2c.

2. Lines 136-137: The loss of receiver chromosomes in the e-Drive configuration vs the split drive configuration could be influenced by the chromosome Cas9 is encoded on. In

the split-drive configuration, the Cas9 was encoded on an autosome whereas the e-Drive encodes it on the X-chromosome. Differences in replication and expression patterns when encoded on sex chromosomes vs autosomes may contribute to higher para-associated non-functional allele-driven lethal mosaicism, that may explain the substantive loss of receiver chromosomes in the e-Drive. Thought experiment, if not a real one: If the Cas9 e-Drive part were encoded on say the ebony locus on the 3rd chromosome, would we still see the same substantive loss of receiver chromosomes, due to lethal mosaicism, at the para locus on the X-chromosome?? Something to ponder upon. I refer to the sex vs autosome debate as I have seen some strange things happen to transgenes (specifically Cas9) when encoded on the X chromosome vs an autosome. Perhaps, future designs should shift to target genes on the autosomes or may be include both as a comparison!! Perhaps include the X vs Autosome differences in the discussion, if feasible.

The reviewer is correct that X-linkage of the eDrive and *vgsc* = *para* locus in *Drosophila* represents a special situation. In other insects the *vgsc* locus is located on an autosome (e.g., Chrom. II in anopheline mosquitoes). In *Drosophila*, placing the eDrive on an autosome (e.g., in *ebony*) would not be predicted to change the dynamics we observe significantly. Since the *yellow* and *para* loci are far apart to undergo virtually free recombination, the eDrive segregates from *para* as though it were located on a different chromosome. However, in other insects where the *vgsc* is located autosomally, one might expect to observe different eDrive dynamics than we find in *Drosophila*. Two notable differences would be expected for an autosomal *vgsc* target allele. On the one hand, loss-of-function *vgsc* alleles generated by action of the eDrive would not be hemizygous lethal in males of the next generation. This factor should delay the culling of lethal alleles compared to the X-linked situation where 50% of lethal *vgsc* alleles are immediately removed from the population. On the other hand, the allelic-drive should take place in both male and female parents for an autosomal *vgsc* target, which should accelerate the drive process. How these two opposing effects would balance is likely to depend quantitatively on various parameters such as the rate of generating NHEJ alleles and copying efficiency of the eDrive element. We have added a short section to the discussion on this interesting point.

3. Lines 147-157: The grandmaternal lineage is generally thought to be associated with higher rates of NHEJ alleles, likely due to cumulative generational maternal Cas9 deposition. It is interesting that Fig.2c & d show relatively higher F->L conversion and lower NHEJ rates in the maternal lineage (F0 Female) vs the F0 male lineage. Could this difference be due to the repair in this case being carried out by HR (DSB) in association with MMR/BER/NER or even TLS followed by MMR/BER/NER. This is possible as the target site at para needs to only repair a mismatch of 2 nucleotides between the donor and receiver chromosomes. HR of the DSB will result in two mismatches, which might be repaired by recruitment of one or more of the aforementioned repair mechanisms.

MMR = MisMatch Repair

BER = Base Excision Repair

NER = Nucleotide Excision Repair

TLS = TransLesion Synthesis

Some thoughts on this would be helpful given the widespread data now available on the so called grandmaternal effect or GESP and will also help expand the discussion around the recruitment of other DNA repair pathways that are likely to be recruited during DSB repair that act in concert with HR. Such a discussion might also strengthen the case of allelic-drive at a distance (reminds me of Einstein's description of quantum entanglement as 'spooky action at a distance') where perhaps DNA repair mechanisms other than HR are recruited leading to higher F->L conversion rates. I think that this discussion is missing from the gene drive literature and should be shed light on.

We believe that the primary factor responsible for the apparent greater F->L conversion in F0 male versus females is due to all NHEJ alleles being 100% lethal in males and only semi-lethal in females (which have two X-chromosomes). The reviewer raises an interesting point regarding the possibility that different repair mechanisms may contribute to repair of DSBs generated in the context of allelic-drive versus cassette copying. In addition to the possibility noted by the reviewer that MMR/BER/NER pathways could potentially contribute to DSB repair in the context of allelic-drive, such repair also would presumably not depend on DNA polymerases that are required for copying gene cassettes. We have added this valid point to the discussion section.

Additionally, in Fig.2d, F2 receiver males, that uncover the allelic conversion rates in the maternal germline, are at 7% (F0 male) and 13% (F0 female) vs the allelic conversion rates in receiver females (Fig.2c) where almost 4 times higher rates are observed (28% and 40%). This is likely the result of both maternal germline conversion as well as zygotic conversion (possibly via MMR, TLS and other repair mechanisms). One could clarify this point.

The difference in this case is actually due to the fact that a significant fraction of target chromosomes fail to be recovered in males where lethal NHEJs are immediately eliminated compared to females where a functional copy can rescue them (thus the ~50% recovery of donor versus receiver chromosomes in females compared to ~2:1 bias in favor of receiver chromosome recovered in males - Fig. 2e). Thus, in F2 males, in which we can directly score receiver chromosome genotypes, we find that 57-63% of surviving target chromosomes are converted. We have clarified this point of potential confusion in the revised text.

A different question that arises is in the FM7 crosses intended to uncover lethal alleles, here maternal germline conversion rates of F to L are at 28% (F0 male) and 20% (F0 female). This is 4x or ~1x higher than in Fig.2d. Is it possible that the para target site in FM7 is accessible to targeting by the e-Drive and subsequent sequencing reveals a higher rate of F to L conversion? An inversion on the balancer chromosome may not curtail the Cas9 RNP from targeting a short 20 bp sequence, no?

The percentages of balanced target chromosomes recovered from F2-females as analyzed in the third generation (shown in Supp. Fig. 3) is approximately on the same order (20-28%) as that in Fig. 2d (28-40%). Given the relatively small numbers of

individual lines established in the three generation crosses we suspect that the modest differences reflect sampling variation.

Lines 147-149, about the ~24% balanced F3 lines not producing any viable males carrying the receiver chromosome, the figure Sup.Fig.3 is a bit confusing to me. If I understand correctly, the 22 and 24% male alleles are actually a % of the number of pair-mated crosses that did not generate balancer-free or receiver chromosome-carrying male progeny. This cannot be sequence-based as one cannot sequence the inviable. If so, how were the 1014L and F alleles frequencies recorded? Are these derived from sequencing the F3 fly lines mentioned in lines 154-157? Were no functional resistant alleles recorded? I am super curious. I think what confused me is the combination of recorded number of crosses with sequencing-based allelic frequencies. Please clarify if I am getting something wrong here as I am a bit confused.

We understand why this seems confusing. As the reviewer correctly infers, loss-of-function NHEJ alleles can only be recovered in females, where they appear to be viable based on approximately equal recovery of donor versus receiver chromosomes in F2 females shown in Fig. 2e,f (left panels). In contrast, in F2 males, there is a significant skew in the percentage of donor versus receiver chromosomes suggesting that a significant fraction of these chromosomes are being either mutated at the target site or damaged in some other fashion (e.g., potentially larger deletions). What we find through sequence analysis of the three generation cross is that ~25% of target chromosomes recovered in females do in fact carry male lethal target site mutations.

The sequence analysis is performed on F3 females carrying both the balancer and derived receiver chromosome that is suspected of carrying the mutation. We use a software program to sort out the WT sequence from the NHEJ sequence to determine what lesion each balanced female carried. This missing information is now provided in the legend to Suppl. Fig.3 and we also added additional information to panels b and c to clarify this analytic process.

Lines 150-151: "female offspring of F3 fly lines." Is my understanding that, female progeny from the F3 balanced lines established and maintained as stocks (progeny of F3 and future generations) were sequenced, accurate?

Yes, we have explained this procedure more fully in the legend to Suppl. Fig. 3 as indicated above.

Line 156: Should it be 20-28% to match Sup.Fig.3b?

Yes, that correction has been made.

Reviewer #3 (Remarks to the Author):

I co-reviewed this manuscript with one of the reviewers who provided the listed reports. This is part of the Nature Communications initiative to facilitate training in peer review and

to provide appropriate recognition for Early Career Researchers who co-review manuscripts.

RESPONSE TO REVIEWERS' COMMENTS

Reviewer #1 (Remarks to the Author):

The authors have responded appropriately to all the comments from both myself and the other reviewers

Reviewer #2 (Remarks to the Author):

The authors have addressed all of my comments in their rebuttal. The only possible omission is the X chromosome vs autosome issue that although they said they have addressed in the discussion of the paper, I couldn't find this in the discussion.

We have added a missing X chromosome vs autosome section to the discussion Line 337-346.

Reviewer #3 (Remarks to the Author):
